# A Killer Disarmed: Natural Killer Cell Impairment in Myelodysplastic Syndrome

**DOI:** 10.3390/cells12040633

**Published:** 2023-02-16

**Authors:** Helena Arellano-Ballestero, May Sabry, Mark W. Lowdell

**Affiliations:** 1Department of Haematology, University College London, London NW3 5PF, UK; 2InmuneBio Inc., Boca Raton, FL 33432, USA; 3Novamune Ltd., London WC2R 1DJ, UK

**Keywords:** acute myeloid leukemia, cancer, immunotherapy, myelodysplastic syndrome, natural killer cells, tumor microenvironment

## Abstract

Myelodysplastic syndrome (MDS) treatment remains a big challenge due to the heterogeneous nature of the disease and its ability to progress to acute myeloid leukemia (AML). The only curative option is allogeneic hematopoietic stem cell transplantation (HSCT), but most patients are unfit for this procedure and are left with only palliative treatment options, causing a big unmet need in the context of this disease. Natural killer (NK) cells are attractive candidates for MDS immunotherapy due to their ability to target myeloid leukemic cells without prior sensitization, and in recent years we have seen an arising number of clinical trials in AML and, recently, MDS. NK cells are reported to be highly dysfunctional in MDS patients, which can be overcome by adoptive NK cell immunotherapy or activation of endogenous NK cells. Here, we review the role of NK cells in MDS, the contribution of the tumor microenvironment (TME) to NK cell impairment, and the most recent data from NK cell-based clinical trials in MDS.

## 1. Introduction

Myelodysplastic syndrome (MDS) is a group of heterogeneous clonal hematopoietic stem cell disorders characterized by cytopenias leading to ineffective hematopoiesis and increased blast production, resulting in bone marrow failure as well as risk of progression to acute myeloid leukemia (AML) [1,2,3]. MDS occurs more frequently in elderly people, which could relate to the increased inflammatory state that is associated with aging, promoting the clonal hematopoiesis that we see in patients [4,5,6]. MDS patients can be classified according to their disease risk; low-risk patients are characterized by bone marrow apoptosis and autoimmune disease manifestations, whereas high-risk patients present immune disfunction, less bone marrow apoptosis, and secondary DNA damage, accelerating the progression to AML [7,8]. On average, 30% of MDS patients progress into AML [9,10,11]. The heterogeneous nature of the disease requires a complex and personalized variety of therapeutic approaches. Among the choices for MDS treatment the only therapy with curative potential is allogenic hematopoietic stem cell transplantation (HSCT), with the remainder of therapeutic approaches involving palliative care. Allogeneic HSCT is associated with a high risk of serious complications such as infections and graft-versus-host disease (GVHD); and since MDS is a disease of the elderly, this option is unavailable to many patients [12]. Currently, the Food and Drug Administration (FDA) has only approved three drugs for the treatment of MDS, the immunomodulatory agent lenalidomide and the hypomethylating agents azacitidine and decitibine. No new drugs have been approved since 2006, and no drug has been approved for second-line treatments such as immunotherapy, occasioning a large unmet need for patients who do not respond to licensed treatments [13].

Natural Killer (NK) cells are attractive candidates for immunotherapy due to their ability to kill cancer cells without prior sensitization. They begin their development in the bone marrow and go through a step-wise process from hematopoietic stem cells (HSCs) to common lymphoid progenitors (CLPs) and NK cell precursors (NKPs) and finally into CD56+ circulating NK cells. CD122 expression whilst in the bone marrow is critical for lineage commitment. NK cells that express CD56 can also be classified according to their maturation status, characterized by the expression levels of CD56. CD56^bright^ NK cells represent a more immature subset that can differentiate into CD56^dim^ NK cells with the acquisition of the Fcγ receptor, CD16 [14,15,16]. NK cells belong to group 1 innate lymphoid cells and make up between 5 to 15% of the total population of circulating lymphocytes [17]. Once activated and triggered, NK cells display direct cytotoxicity against a variety of tumor targets by degranulation; the release of lytic granules containing perforins and granzymes into the synapse between NK and target cell or by induction of apoptosis through the death receptors FasL/TRAIL. Another mechanism is antibody-dependent cell-mediated cytotoxicity (ADCC), where an NK cell will bind to antibodies attached to target cells via the CD16 receptor. Cross-linkage of CD16 leads to perforin and granzyme release causing elimination of the target cell. Triggered NK cells can also secrete cytokines such as tumor necrosis factor-alpha (TNF-α) and interferon-gamma (IFN-γ) leading to a greater influence over immune responses (Figure 1) [18]. In the context of MDS, several studies have reported evidence of NK cell impairment, highlighting the importance of understanding their role in the context of preleukemic myelodysplasia [19,20,21,22]. In this review article, we summarize the role of NK cells in MDS, discuss how the MDS microenvironment influences NK cell function, and highlight the most recent data from clinical trials using different NK cell-based immunotherapeutic approaches for the treatment of this disease.

## 2. NK Cells in MDS

NK cell impairment is frequently reported in cancer to include a reduction in cell numbers, dysregulation of antigen expression, and a display of weaker functional abilities such as cytokine secretion and cytotoxicity. Although earlier studies reported similar NK cell numbers in MDS patients compared to healthy donors [19], more recent reports show that MDS patients exhibit lower numbers of NK cells [22,23,24]. In most cases, this reduction in NK cell number does not correlate with a reduction in T cell lymphocytes. Instead, it is associated with higher risk subgroups of MDS patients according to the WHO and IPSS classification [22]. This is potentially informative, since it points to a particular association between NK cells and disease severity and not a general immunosuppression of broad lymphopenia. It suggests that NK cells are involved in the immune surveillance of dysplastic clones.

The dysregulation of NK cell antigen expression in cancer usually involves the downregulation of activating receptor expression and upregulation of inhibitory receptors, leading to weaker NK cell capacity for target cell recognition and killing. NK cell activation receptors include CD16, which mediates ADCC, and two families of natural cytotoxicity receptors (NCRs): the immunoglobulin superfamily of NKp30, NKp44, NKp46 and those that are C-type lectins, NKG2D and DNAM1. The C-type lectins are particularly important in leukemic cell targeting due to their recognition of MICA/B and ULBP1-6 and PVR (CD155) and Nectin-2 (CD112), respectively [25,26].

A hallmark of NK cells is the presence of killer cell immunoglobulin-like receptors (KIR) that bind to HLA-A/B/C molecules and mostly act as inhibitors of NK cell activation. NK cells also express the CD94/NKG2A complex that recognizes HLA-E and is the main inhibitor of NK activity. The KIR family of genes is highly polymorphic and four major inhibitory KIRs have been defined: KIR2DL2/3, KIR2DL1, KIR3DL1, and KIR3DL2. Immature NK cells will undergo a process called “education” or “licensing” by which they will acquire inhibitory receptors and become mature NK cells; NK cells lacking inhibitory receptors for HLA molecules will become hyporesponsive or anergic and be eliminated. The maturation of NK cells has been described as a gradual process characterized by the loss of NKG2A together with the acquisition of multiple KIR and CD57 [27,28].

Previous studies by Kiladjian et al. investigated NK cell receptor expression in MDS patients and reported comparable expression of NCRs in peripheral blood NK cells of MDS patients versus healthy controls [19]. In a more recent study by Carlsten et al., the downregulation of NKG2D and DNAM-1 expression in bone marrow-derived NK cells from MDS patients was reportedly associated with elevated blast counts and high-risk disease, but similar to Kiladjian et al. they did not see receptor changes in the surface of NK cells from peripheral blood [21]. In contrast, other studies investigating peripheral blood NK cells from MDS patients have reported loss of NKG2D, NKp30, NKp46, CD16, and CD161 expression [20,22,23,24]. Recent studies suggest that NK cell dysfunction in MDS may be attributed to the presence of NK cells with immature phenotypes characterized by an increase in the proportion of CD56^bright^ NK cells, higher “early” KIR expression (KIR2DL2/3), and lower “late” KIR expression (KIR2DL1 and KIR3DL) in MDS patients [22,24,29]. The expression of NK cell ligands in bone marrow aspirates of MDS patients has also been studied since these receptor–ligand interactions are crucial for lysis of MDS blasts, but reports have yielded conflicting results. Epling-Burnette et al. reported that 30% of MDS blasts expressed the NKG2D ligands MICA/B [20]. In another study by Carlsten et al., NKG2D ligands MICs and ULBPs were found to be rarely expressed in the bone marrow of MDS patients. Instead, patients expressed the DNAM1 ligands, CD155 and CD112, showing an important role for DNAM1-CD155/CD112 interactions in the killing of MDS blasts [21].

These conflicting data on NK cell receptor expression in MDS may be attributed to the source of NK cells, whether obtained from peripheral blood or bone marrow, and timing of samples especially as it relates to the administration of chemotherapeutic agents, which are known to have an effect on the state of NK cells [30,31,32]. Taken together, these studies demonstrate an NK cell phenotype characterized by lack of maturation markers and the downregulation of NK cell activation receptors in MDS patients, which is associated with NK cell functional impairment.

Several studies have reported impaired or dysfunctional NK cell cytokine secretion or cytotoxicity in MDS patients. A reduction in MDS-NK cell secretion of TNF-α and IFN-γ was previously demonstrated in response to IL-2 and K562 stimulation [19,22]. NK cell cytotoxicity through perforin and granzyme release is associated with enhanced degranulation as measured by LAMP-1/CD107a expression or by the activation of death receptor-mediated pathways TRAIL and FasL [33,34]. Carlsten et al. reported lower degranulation in MDS-NK cells following exposure to K562 leukemic target cells relative to healthy controls [21]. Hejazi et al. showed no difference in the mobilization of CD107a, a molecule expressed on the NK cell surface after degranulation, when MDS-NK cells were exposed to K562. However, they showed a significant decrease in MDS-NK cell killing activity without any associated changes in CD107a expression compared to healthy donors. This observation prompted them to investigate whether degranulating MDS-NK cells are properly armed with cytotoxic molecules. They observed a substantial reduction in perforin and granzyme B loading of granules in MDS-NK cells, which explains why NK cells might still be able to mobilize CD107a to the cell surface without effective killing of target cells [22]. Similarly, other studies have shown impaired lysis of K562 and CD34^+^ blasts by NK cells from MDS patients [19,20,21,22].

In a recent study, Tsirogianni et al. [35] analyzed peripheral blood NK cells from MDS patients who had been treated with azacytidine, and they tested their ability to lyse K562 leukemic target cells in vitro. Patients with higher lytic function showed significantly longer overall survival. Indeed, the association was so strong that the group was able to calculate a threshold of NK mediated lysis, which was predictive for survival beyond 2 years. Patients below the threshold showed a median overall survival of 18 months compared to those falling above the threshold with a median survival of 52 months. Collectively, these studies suggest that NK cell antitumor functions may be critical in the response to MDS. Their impairment, however, can be influenced by numerous factors in the tumor microenvironment that contribute to disease progression.

## 3. The Tumor Microenvironment in MDS

There is growing evidence that different components of the tumor microenvironment (TME), including other immune cells and soluble factors, contribute to NK cell impairment in MDS (Figure 2) although whether the TME is responsible for the dysfunction of NK cells isolated from the peripheral blood of MDS and other cancer patients is unknown. One possibility is that the circulating NK cells isolated from blood have trafficked through the TME and suffered long term impairment. However, Imai et al. [36] showed elegantly that healthy individuals show a range of NK cell functional abilities, and those with lower NK function have a higher lifetime risk of cancer. This supports the notion that NK cells mediate tumor immune surveillance and the first stages of impaired function occur prior to the development of cancer and a TME. So, passage through a TME is not required for impairment of NK activity, but the composition of the TME can certainly provide additional suppression of NK cell function.

Dendritic cells (DCs) in MDS patients have been reported to have an immature phenotype that is observed on a transcriptional and functional level [37,38]; this will have negative effects on DC-NK cell crosstalk and the release of activating soluble factors such as IL-12, IL-18, and type-I IFN, which prime NK cells for tumor lysis [39]. Increased levels of vascular endothelial growth factor (VEGF) have been found in the serum of MDS patients, which will have a series of effects in the TME [40,41]; increased VEGF will contribute to the inhibition of DC maturation [42,43] and induction of T regulatory cell (Treg) activation, [42,43], which is associated with an unfavorable disease prognosis in MDS [8,44,45,46,47,48]. Tregs can play two opposing roles in MDS; in the low-risk group Tregs appear to be dysfunctional, favoring the selection of dysplastic clones; in the high-risk group, Treg numbers are expanded and can suppress immune responses against transformed clones, promoting leukemia progression [8,44,45,46,47,48]. Tregs are well known to suppress NK cell function via cell-to-cell contact or through the secretion of soluble factors such as IL-10 or TGF-β [49,50]. Furthermore, in vivo murine models of AML have shown that Treg depletion allows better NK cell activation [51]. 

Another source of VEGF in MDS is from myeloid-derived suppressor cells (MDSCs) [52], which are expanded in the bone marrow of MDS patients and correlate with high-risk disease [24,52,53,54]. Two mechanisms have been described on how MDSCs are recruited into the bone marrow of MDS patients. The inflammatory environment in the MDS bone marrow contains inflammation associated signaling molecules, such as the danger-associated molecular pattern (DAMP) heterodimer S100A8/S100A9 that will interact with its ligand CD33 on MDSCs. S100A8/S100A9 has been reported to be overexpressed in MDS, allowing its accumulation and activation [52]. Another mechanism that has been described is via the CXCR4/CXCL12 axis, where the stromal cells in the bone marrow niche supporting MDS blasts have been shown to overexpress CXCL12. The MDSCs in these patients show an upregulation of CXCR4, a ligand for CXCL12 [54,55]. CXCL12 is an important chemokine for MDSCs, and its hyperexpression is thought to mediate the increased frequency of MDSCs in the bone marrow of MDS patients.

Furthermore, MDSCs in MDS have been described as hyperfunctional and have increased suppressive activity by secreting IL-10, TGF-β, and reactive oxygen species (ROS) that will ultimately impair NK cells [52,56,57]. In chronic myeloid leukemia, AML, and chronic myelomonocytic leukemia it has been shown that ROS triggers apoptosis of NK cells in the TME and also induce downregulation of activating receptors, leading to dysfunctional NK cells [58,59,60]. Another source of ROS is blood transfusions that MDS patients receive as part of their treatment. These often induce iron overload, causing ROS generation and therefore NK cell impairment [61,62,63].

Mesenchymal stromal cells (MSCs) are another cell type that is reportedly impaired in MDS patients. These cells exhibit reduced proliferative capacity and poor ability to differentiate into other cell types; altered morphology; and alteration of molecules and cytokines involved in hematopoietic stem and progenitor cells (HSPCs)–MSCs interactions [64,65,66]. MSCs have a very important role in supporting HSPCs in their renewal and differentiation, which will allow the production of mature blood cells, including NK cells [67,68].

Hejazi and colleagues argue in their paper that the absence of stromal support in the microenvironment in MDS patients could be leading to the lack of maturity in patient NK cells, as it is known that full maturation of NK cells that express KIR ligands requires the presence of stromal cells [22,69]. The increased secretion of the pro-inflammatory cytokine IL-6 by MSCs has also been reported in MDS [54,70], contributing to general inflammation in the bone marrow of patients. This inflammatory state in the bone marrow of MDS patients has been described by increased levels of pro-inflammatory cytokines such as IFN-γ and TNF-α, which will induce apoptosis on CD34 cells and bone marrow failure, helping in the selection of malignant CD34 cells [71,72]. Importantly, TNF-α activates the transcription factor NF-κβ, a regulator of cell signaling, proliferation, and differentiation [73]. NF-κβ levels are increased in high-risk MDS patients, corresponding with an increase of blast counts [74]. Furthermore, some studies have shown that NF-κβ signaling contributes to leukemia progression [75], which could play an important role also in transition of MDS to AML. 

Hypoxia is another element that has been described to have an effect in the bone marrow of MDS patients. Hypoxic conditions have been shown to select CD34 malignant cells with stem cell potential in MDS patients [76]. Expanded CD34 malignant blasts will further contribute to inflammation in the bone marrow by secreting pro-inflammatory cytokines such as IL-8 and TGF-β into the TME [77,78,79,80,81,82], increasing levels of ROS [83,84,85], and MSCs inhibition via TGF-β secretion [82,86], ultimately affecting NK cells.

As we have seen, a broad range of different elements in the TME of MDS patients appear to be active, all of which may compromise NK cell interaction with other cell types or affect NK cells indirectly through secretion of inhibitory cytokines. Understanding the role of the TME and how it contributes to NK cell impairment is crucial in order to develop successful anticancer therapies that could include combination strategies targeting different elements of the TME, thus enhancing NK cell activity for successful blast killing. 

## 4. NK Cell Immunotherapy in MDS

Different immunotherapeutic strategies for NK cell modulation have been developed for MDS treatment. Preclinical studies have reported enhanced MDS patient-derived NK cell proliferation, granule secretion, and cytotoxicity after exposure to IL-2, showing that NK cell dysfunction can be overcome [20,22]. The only curative option nowadays for patients with MDS is allogeneic HSCT from an HLA-identical donor. However, one of the main problems is the recurrence of underlying leukemia after the transplant, which is usually the cause of treatment failure [87]. Therefore, patients who are refractory or have relapsed are left with few options. In these cases, NK cell immunotherapy, which has shown positive results both in solid and in liquid cancers (reviewed here [88]), could be a suitable option. In the context of MDS, NK cells have been tested clinically in the past, and there are several ongoing clinical trials at the time of writing (Table 1). Two different options have been considered; the first one involves adoptive immunotherapy using allogeneic donor NK from a variety of sources: adult peripheral blood, umbilical cord blood, and hematopoietic progenitor cells from iPSC cells. The second approach has been to potentiate the patients’ own NK cells in vivo.

The infusion of NK cells from an HLA matched or unmatched donor is the most common approach, but, in most cases, this requires the infusion of cytokines such as IL-2 or IL-15 to expand them and sustain them in vivo. Some trials using IL-2 activated adoptive NK cell therapy have shown good results in patients with relapsed or refractory AML, with complete remissions in up to 50% of the patients [89]. Even though it is a widely used strategy, cytokine infusions have previously been shown to induce severe toxic side effects such as cytokine release syndrome, and the development of novel strategies that sustain NK cell proliferation and activation in vivo without toxicity will be important moving forward. Low-dose IL-2 is also known to induce Treg activation [90]. Tregs have a higher affinity for IL-2, and thus there is a great risk of starving NK of IL-2 whist simultaneously activating Treg to further suppress NK cell function [91]. The administration of donor NK cells together with IL-2 and with the TGF-β receptor I inhibitor, vactosertib, is being trialed in relapsed patients with solid and liquid cancers, including MDS (NCT05400122). It has been shown that TGF-β inhibitors such as vactosertib can help attenuate Tregs [92,93]; hence, this combination of therapies holds some promise in reducing unwanted side effects of cytokine administration.

IL-15 is another cytokine that has been used in the clinical setting, but as opposed to IL-2, IL-15 holds some advantages such as its superior role in the stimulation of NK cells and T cells and also the fact that it does not stimulate Treg proliferation [94,95]. Cooley et al. performed the first clinical trial in AML/MDS with infusion of haploidentical NK cells with IL-15 doses afterwards. Patients received either intravenous (NCT01385423) or subcutaneous (NCT02395822) IL-15 doses after NK cell infusion. In phase I, 36% of patients had robust in vivo NK cell expansion after 14 days, and 32% achieved complete remission. In phase II, 27% of patients had NK cell expansion after 14 days, and 40% achieved complete remission. IL-15 infusion induced better in vivo expansion than previous trials with IL-2, but it was associated with cytokine release syndrome in 56% of patients when given subcutaneously but not intravenously [96]. At the time of writing this manuscript, there is an ongoing trial (NCT02890758) for different types of cancers, including MDS, where patients receive NK cells from HLA-mismatched donors either alone or in combination with IL-15 infusions.

A recent development in NK cell biology is the concept of NK cells activated in vivo or ex vivo such that they obtain a memory-like phenotype (ml-NK). These ml-NKs can be generated by short-term exposure to a combination of IL-12, IL-15, and IL-18, in which case they are commonly termed “cytokine-induced ml-NK”—CIML-NK [97,98,99]. CIML-NK show better capacity to respond after a second activation with cytokines or by engagement of activating NK receptors. The first report of clinical use of CIML-NK was in patients with MDS and AML [97]. Haploidentical NK cells from donors were pre-activated for 12-16 h with IL-12, IL-15, and IL-18 ex vivo and then infused into patients who had been preconditioned with lymphodepleting chemotherapy. The infusions were supplemented with low dose IL-2. These CIML-NK were not inhibited by KIR/KIR-ligand interactions and were highly lytic to myeloid leukemic cells in vitro. Moreover, the infused NK cells expanded in vivo were associated with clinical responses in over half of the patients treated, including complete remissions in four or five patients who responded. This phase I trial transitioned to a phase II trial in which the low dose, subcutaneous IL-2 infusions were replaced with IL-15 (NCT01898793), but the trial was terminated due to insufficient funding/staff. Another trial in the use of CIML in hematologic malignancies including juvenile myelomonocytic leukemia is also ongoing (NCT04024761). In this trial, CIML NK cells will be generated and infused to the patients, but no cytokine infusions will be given afterwards. As seen here, the use of cytokines to expand and activate NK cells from patients in vivo or ex vivo has been widely used in the clinical setting showing some promising results, but cytokine release syndrome is a common and severe side effect that should not go unnoticed. Ex vivo incubation of NK cells with cytokines has achieved greater expansion numbers and activation of cells; however, cytokine infusions in the patients are still needed afterwards in order to sustain NK cells. Strategies to activate and expand NK cells that do not include cytokines should be further explored, and we may see an increase of other clinical trials in the upcoming years.

The use of feeder-cell ex vivo is another alternative for the expansion and activation of NK cells. In vitro data from the use of the irradiated K562 cell line modified to express membrane bound IL-15 and 41BB ligand (K562-mbIL15-41BBL) has shown a 26-fold expansion of NK cells and powerful antileukemic activity, with the ability to eradicate AML in mouse models [100,101,102]. Better in vitro expansion numbers have been achieved with the use of K562-mbIL21 where NK cells were expanded by up to 35,000-fold in vitro [103,104,105]. At this moment, there is an ongoing clinical trial using expanded haploidentical NK cells with K562-mbIL21 (NCT04220684).

Chimeric antigen receptor (CAR)-T cells have shown remarkable results in some blood cancers such as acute lymphoblastic leukemia, chronic lymphocytic leukemia, and lymphoma [106,107,108] and have also been used in clinical trials for AML/MDS [109,110,111,112,113]. However, despite their success in the clinics, autologous CAR-T cells present some limitations related to the fact that these drugs are personalized to only one patient, making them complex and expensive. Furthermore, in some patients CAR-T cell treatment has been linked to cytokine release syndrome and neurotoxicity [114]. Allogeneic CAR-NK cells are considered potential alternatives to autologous CAR-T since allogeneic NK cells do not induce graft-versus-host disease and may be available as an off-the-shelf alternative. NK cells can be sourced from an allogeneic donor without the need of full HLA matching; hence, they do not need to be produced individually for each patient. They also appear to be safe when infused to patients with cancer [115,116,117]. Nonetheless, one of the main challenges to develop CAR therapies for MDS/AML remains the lack of specific ligands, which are usually expressed in normal myeloid cells [118,119]. Tang et al. performed the first clinical trial using CAR-NK cells in AML, targeting CD33 and using the NK92 cell line as the source of NK cells. They reported no significant adverse effects and concluded that, even though they did not see any clinical effect, CAR-NK cells were safe to administer [120]. Several other molecules are being used in CAR-NK therapy in clinical trials in leukemia, but no results have been posted so far: CD7 (NCT02742727), CD19 (NCT02892695, NCT03056339, NCT04796675, NCT00995137) and NKG2DL (NCT04623944). There are other preclinical studies looking at the use of CAR-NK cells in AML. Sinha et al. studied a CAR construct with CD123 in NK92 cells and evaluated their efficacy; they observed how these cells exhibited higher cytotoxicity against AML cell lines expressing CD123 both in vitro and in vivo models [121]. Dong et al. engineered memory-like NK cells to express a CAR against NPM1c, an antigen expressed 35% in AML that showed enhanced cytotoxicity in in vivo models [122]. There are no current data on the use of CAR-NK cells for MDS patients, but some trials are ongoing using allogeneic CAR-NK cells targeting NKG2D ligands (NCT04623944) and CAR.70/IL15-transduced NK cells derived from cord blood (NCT05092451). The use of CAR-NK cells offers an exciting opportunity to develop an off-the-shelf product, but identifying markers that can target MDS blasts exclusively are crucial and still remain one of the biggest challenges.

The in vivo potentiation of the patient’s own NK cells is another strategy that has been considered. Bispecific and trispecific killer cell engagers (BiKE and TriKE) are single-chain variable fragment recombinant reagents that contain an anti-CD16 expressed on effector NK cells and other antigens of interest in cancer cells. Gleason et al. tested the ability of a BiKE for CD16 and CD33 to induce NK cell function in MDS patients in vitro, where CD33 engages myeloid targets including MDS. Their study showed enhanced degranulation and TNF-α and INF-γ secretion in NK cells from patients [23]. Other TriKEs have been studied in the context of MDS, where besides the anti-CD16 and the anti-CD33, a modified IL-15 linker was added in order to induce NK cell proliferation. These studies reported enhanced killing kinetics and evasion of MDSCs suppression [123,124]. At the time of writing this manuscript, there was an ongoing trial (NCT03214666) testing the TriKE GTB-3550 in MDS and other hematological malignancies, but it was terminated due to the development of the second generation TriKE GTB-3650. Overall, BiKEs and TriKEs have shown promising results both in vitro and in vivo and provide a versatile off-the-shelf option; however, safety concerns remain since IL-15 stimulation of inflammatory T cells could trigger cytokine release syndrome. 

The activation of NK cells has also been studied in the context of tumor-priming. NK cells that are exposed to the ALL cell line INB16 enter a primed stage where they are ready to lyse upon exposure to a second tumor cell signal, even if the second signals arise from NK-resistant tumor cells [125,126,127,128]. Tumor-primed NK cells have also been used in clinical trials for AML/MDS, with little toxicity and some sustained clinical responses. This is of particular note since these were the first clinical trials of an adoptive NK therapy in the absence of cytokine support. Allogeneic NK cells from related, haploidentical donors were incubated overnight with a lysate of INB16 that was then removed, and the tumor-primed NK cells were cryopreserved, shipped, and infused into the patients one day after completion of lymphoreductive chemotherapy [129]. Four of the seven patients who completed treatment showed clinical responses with one patient with chemo-resistant AML achieving CR, which was sustained for over 11 months. In a second trial, 12 AML patients at high risk of recurrence were treated following the same protocol, where two patients remained relapse-free in posttrial follow-up, and three patients remained in complete remission for 32.6 to 47.6+ months [130]. 

The observation that MDS patients with moderate NK cell lytic function have a significantly better prognosis than those with low or undetectable NK medicated killing of K562 cells [32] led to the hypothesis that activation of endogenous NK cells in those MDS patients with low NK function might move them above the threshold for improved overall survival. Tumor-priming of endogenous NK cells in vivo is being trialed at the moment in high-risk MDS and AML patients (EudraCT 2019-004820-40) who receive three weekly infusions of a replication-incompetent preparation of the INB16 cell line. These patients receive no lymphoreductive conditioning chemotherapy and no cytokine support, resulting in a treatment that is well tolerated and compatible with an elderly patient cohort. 

**Table 1 cells-12-00633-t001:** Summary of NK cell-based clinical trials for MDS.

Approach/Treatment	Trial Phase	Status	Patient Numbers	Trial Identifier	References
Adoptive NK cell immunotherapy					
+cytokine stimulation with IL-2/-15/CIML	Phase I	Completed	26	NCT01385423	[96]
		Ongoing	76	NCT05400122, NCT02890758, NCT04024761	[92,93]
	Phase II	Completed	17	NCT02395822	[96]
	Phase I/II	Terminated	89	NCT01898793	[97,98,99]
+expanded K562-mbIL21	Phase I	Ongoing	30	NCT04220684	[100,101,102,103,104,105]
+CAR NK cells	Phase I	Ongoing	90	NCT04623944	[120,121,122]
	Phase I/II	Ongoing	94	NCT05092451	
In vivo NK cell activation					
BiKE/TriKE	Phase I/II	Terminated	12	NCT03214666	[23,123,124]
INB16 priming	Phase I	Ongoing	12	EudraCT 2019-004820-40	[125,126,127,128,129,130]

Abbreviations: IL, interleukin; CIML, cytokine-induced memory-like; mbIL21. membrane bound IL-21; CAR, chimeric antigen receptor; BiKE/TriKE, bi/trispecific killer cell engagers.

To summarize, NK cell immunotherapy in AML/MDS offers a wide range of options that show exciting opportunities and appear to be safe for the treatment of this disease, but relapse rates are still high, and side effects of concurrent cytokine treatment are common. Cytokine administration is usually a requirement in order to sustain adoptively transferred NK cells, especially those generated by extensive ex vivo expansion. The development of better strategies for NK cell expansion and persistence post infusion remains a central tenet of research, including transduction of mbIL-15 constructs into the NK product. The use of feeder-cells to increase expansion of mature NK, or iPSC as a reliable source of proliferative NK cells, promise to reduce the cost of manufacture of allogeneic NK cells for adoptive immunotherapy. This is allowing the use of multiple NK infusions and is a move away from the concept of “NK engraftment and cure” towards one of “NK injection and control” where NK cell administrations become an ongoing treatment. The success of this strategy is predicated on the assumption that repeat infusions of allogeneic NK cells will not require lymphodepleting chemotherapy before each injection, and this is being tested in trials currently. The need for adoptive NK immunotherapy may be avoided by enhancing endogenous NK function, and BiKEs/TriKEs and in vivo generation of ml-NK cells are exciting off-the-shelf products that have shown interesting results that could benefit many patients as they are relatively inexpensive, are not personalized, and appear to be well tolerated, which might result in a surge of NK cell therapies in the upcoming years.

## 5. Conclusions

The heterogeneous nature of MDS hinders treatment of this disease, where delaying progression to secondary lethal AML is one of the key challenges in managing patients. NK cells play a crucial role in immune surveillance, and their impairment in the context of MDS suggests that is important to understand their role in this disease. The presence of a highly suppressive TME further impairs NK cell function, making it equally important to understand. Nonetheless, exciting in vitro and in vivo data suggest that NK cell impairment can be overcome in MDS, and results from early clinical trials are encouraging. One of the main challenges remains the presence of low numbers of NK cells seen in some MDS patients and the consequent need for adoptive immunotherapy. Challenges remain to sustain adoptive NK cells in vivo, although the use of cytokines remains a common yet problematic strategy, or to develop allogeneic NK cells that do not provoke allo-immunity, and thus repeat NK doses can be given. The imatinib family of drugs has revolutionized the treatment of chronic myeloid leukemia so that patients are no longer exposed to the high risk of allogeneic hematopoietic stem cell transplants but are maintained disease-free with repeat treatments. The very minimal toxicity of allogeneic NK cell infusions suggests that these too might become routine treatments to control disease. The field of NK cell immunotherapy is still expanding and holds a lot of potential to be unraveled over the coming years, leading to yet more novel therapies and the chance for significant patient benefit.

## Figures and Tables

**Figure 1 cells-12-00633-f001:**
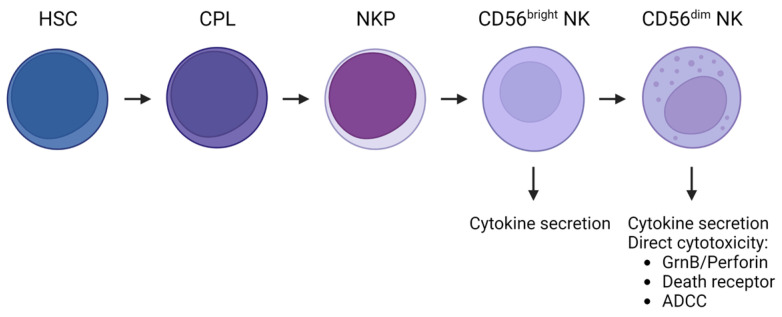
Stages of NK cell development and function. Schematic representation of the different stages of NK cell differentiation and maturation in humans. HSCs differentiate into CPLs and then into NKPs. The acquisition of CD56 will determine the last step into NK cells. CD56^bright^ NK cells represent a more immature subset that can develop into CD56^dim^ NK cells. Both subsets are capable of cytokine secretion, but CD56^dim^ NK cells are more cytotoxic. Direct cytotoxicity can be exerted via granule secretion, death receptor ligation, or ADCC.

**Figure 2 cells-12-00633-f002:**
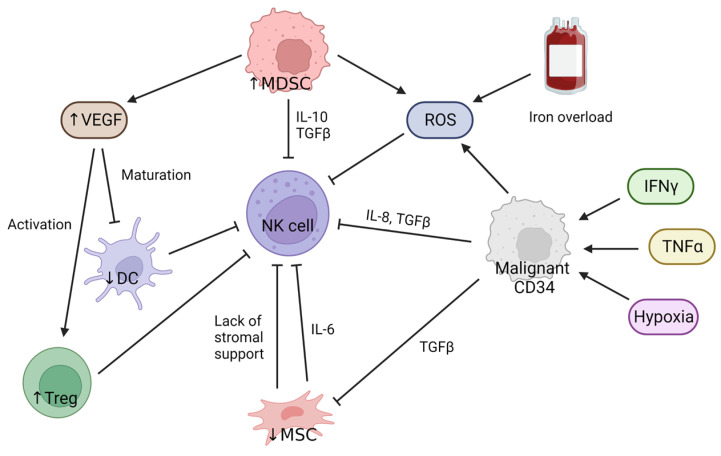
MDS impairs NK cell function through TME factors. The presence of immature DCs directly affects NK cell activation. Increased levels of VEGF contribute to DC immaturity and also Treg expansion. Increased levels of hyperactive Tregs secrete IL-10 and TGF-β into the TME, suppressing NK cell activation. A source of VEGF in MDS are MDSCs, which are expanded and hyperactive and contribute to NK cell impairment by secreting IL-10, TGF-β, and ROS into the TME. Another source of ROS are blood transfusions received by MDS patients as part of their treatment. MSCs are also reported to be reduced in numbers and have an impaired phenotype in MDS; impaired MSCs are not able to support NK cell maturation and contribute to general inflammation by secreting pro-inflammatory cytokines such as IL-6. The general inflammatory state in the bone marrow of MDS patients is characterized by an increase of IFN-γ and TNF-α and increased hypoxia, inducing bone marrow apoptosis that helps in the selection of malignant CD34 cells. Those cells further secrete ROS and pro-inflammatory cytokines such as IL-8 and TGF-β, impairing directly or indirectly NK cell function and activation.

## Data Availability

Not applicable.

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
