# Peer review of "A Killer Disarmed: Natural Killer Cell Impairment in Myelodysplastic Syndrome"

_cells, 2023, doi:10.3390/cells12040633_

Round 1

Reviewer 1 Report

This review summarises recent research and clinical trials of Natural Killer (NK) cell-based immunotherapy to treat myelodysplastic syndrome. The authors discuss the various approaches that are being taken to induce NK activity against malignant cells. This is a well written and informative review that proveds a good summary of the various approaches to using NK cells to treat MDS.

The review would be improved by updating the infomation relating to the cited clinical trials. Some that are cited as ongoing have been completed or terminated eg NCT01898793 and NCT03214666.

Author Response

Thank you for reviewing our manuscript and for your helpful suggestion regarding updating the outcome of the trials we cited. In fact, we have presented all of the outcome data which are in the public domain at present and those trials which have completed or were terminated have not submitted data nor published their findings as far as we can ascertain. We will continue to monitor the literature prior to finalisation of the manuscript and append any data which arise.

Reviewer 2 Report

The authors focused on NK cell dysfunction in MDS and discussed its status and causes in the literature.

The description is well written, and the review of a series of papers with conflicting results from the past to the present is a nice touch.

In particular, it is very reasonable and understandable that TME-induced NK cell dysfunction occurs in MDS cases, as shown in Figure 1.

However, even if the apparent dysfunction of NK cells is caused by such environmental changes, if the NK cells are transferred to in vitro and their cell-killing ability is experimentally examined by targeting K562 cells, etc., they will probably show the same function as NK cells from healthy individuals if the environment is favorable. If the environment is favorable, they may function in the same way as NK cells derived from healthy individuals.

If not, irreversible changes in genes or signal transduction may occur (by TME?). If this is not the case, does it mean that irreversible changes in genes or signal transduction are occurring (by TME?)?

Please consider these points.

Author Response

Thank you for reviewing our paper. Your point regarding NK dysfunction in the TME versus NK cells tested in vitro is interesting. The published literature largely reports that NK cells from peripheral blood of cancer patients are dysfunctional in conventional in vitro assays such as K562 lysis compared to NK cells from healthy donors. These cells are not usually isolated from the TME and thus the impairment is systemic; it is unknown whether these peripheral blood NK cells have passaged through the TME or whether NK dysfunction is constitutive to the individual (as reported by Irai et al 2000) and pre-determines the subject's risk to cancer. We will include a paragraph in the Discussion regarding possible mechanisms behind the peripheral blood NK impairment as you suggest.

Reviewer 3 Report

In this paper, the Authors review the function of NK cells in pathogenesis of MDS, evaluating also the potential therapeutic implications and the ongoing clinical trial involving NK cells in MDS.

The review is overall comprehensive and well written and allows the reader to easily understand the role of NK cells in this setting.

Minor:

- A figure resuming the main mechanism of killing of NK cells as well as modification during maturation may further facilitate the reading of the review

- In the conclusion session, the Authors state that the imatinib family of drugs is not able to cure CML. This is not completely true as in recent clinical trials many patients in continuous deep molecular remission stopped TKI administration and remained disease free. I think that the treatment of CLL with BTK inhibitors represents a better analogy for chronic NK-based treatment, as BTK targeting treatment is continuously delivered until progression and treatment-free remission is significantly less likely in this setting.

Author Response

Thank you for reviewing our manuscript. You are quite correct that some CML patients are now being reported to remain in CR after cessation of TKI and we have amended that comment. The suggestion to discuss CLL patients treated with BTK inhibitors is excellent and we will incorporate that analogy.